# Novel Design of a Multimodal Technology-Based Smart Stethoscope for Personal Cardiovascular Health Monitoring

**DOI:** 10.3390/s22176465

**Published:** 2022-08-27

**Authors:** Heejoon Park, Qun Wei, Soomin Lee, Miran Lee

**Affiliations:** 1Department of Biomedical Engineering, School of Medicine, Keimyung University, Daegu 42601, Korea; 2Clairaudience Company Limited, Daegu 42403, Korea; 3Department of Biomedical Engineering, Graduate School of Medicine, Keimyung University, Daegu 42601, Korea; 4Department of Computer Information & Engineering, Daegu University, Daegu 38453, Korea

**Keywords:** multimodal, heart sound, phonocardiogram (PCG), pulse, photoplethysmogram (PPG), smart stethoscope, cardiovascular health, usability

## Abstract

Heart sounds and heart rate (pulse) are the most common physiological signals used in the diagnosis of cardiovascular diseases. Measuring these signals using a device and analyzing their interrelationships simultaneously can improve the accuracy of existing methods and propose new approaches for the diagnosis of cardiovascular diseases. In this study, we have presented a novel smart stethoscope based on multimodal physiological signal measurement technology for personal cardiovascular health monitoring. The proposed device is designed in the shape of a compact personal computer mouse for easy grasping and attachment to the surface of the chest using only one hand. A digital microphone and photoplehysmogram sensor are installed on the bottom and top surfaces of the device, respectively, to measure heart sound and pulse from the user’s chest and finger simultaneously. In addition, a high-performance Bluetooth Low Energy System-on-Chip ARM microprocessor is used for pre-processing of measured data and communication with the smartphone. The prototype is assembled on a manufactured printed circuit board and 3D-printed shell to conduct an in vivo experiment to test the performance of physiological signal measurement and usability by observing users’ muscle fatigue variation.

## 1. Introduction

Heart sound auscultation, electrocardiogram (ECG) recording, and heart rate measurement are the most popular methods for observing and analyzing the cardiovascular status in in vitro diagnostics [1]. Each technique represents different mechanisms when blood flows from and back to the heart through the circulatory system, such as cardiac motion, blood flow, and blood pressure [2]. Owing to the rapid development of electrical engineering technology, the use of portable digital health monitoring devices for personal cardiovascular health monitoring has increased rapidly over the last two decades. The devices can be grossly divided into two categories: those for medical use, such as digital stethoscopes and 24 h Holter ECG systems, and wearable devices, such as health bands and smartwatches. In a digital stethoscope, the traditional acoustic transducer is replaced with a digital microphone sensor inside the bell-shaped acoustic stethoscope. It feeds signals to an analog-to-digital converter for heart sound acquisition [3]. The design includes preamplification and filter circuits to amplify heart sounds and improve the signal quality. The 24 h Holter ECG system is a battery-packaged device that is easy to carry and can record the ECG signals for 24 h. Generally, Holters have more than five leads with AgCl electrodes attached to the skin’s surface around the chest for ECG signal measurement [4]. It is well known that these two methods are widely used in screening cardiovascular diseases because of their small size, ease of use, and relatively low price compared to the professional equipment used in patient monitoring, X-ray, and computer tomography (CT). However, cardiovascular diseases cannot be diagnosed by heart sounds and ECG signals alone because of the weak intensity of these two types of signals frequently combined with a variety of noise signals. In addition, users without prior medical knowledge find it difficult to understand and analyze the raw data of the heart sound and ECG signals acquired by digital stethoscopes and Holters.

Recently, with the popularity of wearable devices in personal management, the use of health bands and smartwatches to record the daily health status of cardiovascular system by measuring pulse and ECG from the user’s wrist is widely available. The pulse signal is measured by a photoplethysmogram (PPG) sensor commonly designed at the bottom of the health band or smartwatches and attached to the skin surface of the wrist. The measured signal is processed and delineated to the heart rate to observe the heart pump and blood flow status [5]. To measure ECG signals, a two-electrode ECG measurement technology has been applied to the smartwatch: one electrode is placed at the bottom of the watch that is attached to the skin, and the other is always placed around the watch’s case to allow the user to use another hand to touch the watch, thus forming a circle of the body to conduct current from the heart to the watch for ECG signal measurement [6]. These two methods are suitable for observing heart status in daily life and providing an alarm during an emergency [7,8]. However, a common disadvantage of these two methods in wearable devices is that the measured data are discrete because the user cannot maintain the same gesture during signal measurement for a long time; the value of heart rate is calculated by the signal measured for 10 s and multiplied by 6 to obtain the rates in one minute. In addition, the ECG signal is measured by the smartwatch only when the user touches the watch with both hands.

The cardiovascular system is the most complex system in the human body, that includes the heart, blood vessels, and blood. However, existing personal cardiovascular monitoring devices, such as stethoscopes, Holter, and health bands, can measure only one type of signal from the cardiovascular system. Therefore, for cardiovascular disease diagnosis, the acquired data from these devices can only be used as a reference for initial screening and are not suitable for diagnostics. To improve the accuracy of existing methods and identify novel approaches for the diagnosis of cardiovascular diseases, wearable devices based on multimodal technology were developed for measuring heart sound, ECG, and pulse simultaneously; however, analysis of their interrelationship has been reported in several studies [9]. Kulm et al. presented a heart sound and ECG signal acquisition device using a multimodal ECG system along with a stethoscope at the back of the device for simultaneous data recording of the heart. However, the device has to be fixed on the skin of the chest surface using Ag-cl electrodes and size of the device without a battery is 60 mm × 70 mm × 6 mm, which is not suitable for daily use [10]. One of our previous studies also presented a multimodal technology-based smart stethoscope with a folding finger-ring shape design to simultaneously measure heart sound and pulse signals [11]. Owing to the finger-ring shape, the developed device can be worn on the user’s finger and placed on the chest to measure heart sounds and PPG signals simultaneously. This is convenient and suitable for daily use. However, the circuit of the device was separately mounted on two printed circuit boards (PCBs) due to the folding structure design, which resulted in the breakage of connection cables when the device was folded repeatedly for opening and closing. In addition, the size of the finger hole was fixed; therefore, it was not suitable for all users.

Based on previous research, we have presented a novel multimodal technology-based smart stethoscope designed for personal cardiovascular health monitoring. To improve convenience, mobility, and durability, the proposed smart stethoscope is designed in the shape of a compact personal computer (PC) mouse that can be grasped by one hand and attached to the user’s chest easily. The stethoscope bell and PPG sensor are attached to the bottom and top of the device, respectively, to simultaneously measure the heart sound and pulse from the surface of the user’s chest and finger, respectively. In addition, a high-performance Bluetooth Low Energy (BLE) System-on-Chip (SoC) ARM processor is implemented in the device for system control, data pre-processing, and wireless data communication. The prototype is used to conduct an in vivo experiment with 20 young adults to verify the performance of physiological signal measurement and test the usability of the device by evaluating muscle fatigue levels.

## 2. Methods

### 2.1. Design of the Shape of Proposed Smart Stethoscope

The goal of this study is to develop a smart stethoscope that can be held by one hand to measure heart sounds and pulses simultaneously. Therefore, the reference for the shape of the proposed smart stethoscope was taken from a standard PC mouse because of its perfect grip and appropriate size suitable for use by every individual [12]. Figure 1a shows the design of the shape of proposed multimodal technology-based smart stethoscope for personal cardiovascular status monitoring. The size of the device is set as 10 × 5 × 3 cm, which is the standard size of a PC mouse, so that it can be held comfortably with on hand. In addition, a folding finger holder similar to a smartphone ring grip is provided because the users have to hold the device and attach it to their chest by themselves. It is designed at the center of the device so that it is easy to fasten the finger during time-consuming measurements. For pulse signal measurement, a hole is made in the middle of the PPG sensor to closely approach the first segment of the finger (distal phalange). When the user inserts the finger into the finger holder to grasp the device, the finger is guided onto the PPG sensor hole to measure the pulse directly. In addition, three push switches are designed under the finger hole for on/off device power and mode selection.

Figure 1b shows the bottom of the device, which includes a stethoscope bell and display screen for heart-sound measurement and information display, respectively. The stethoscope bell is positioned on top of the bottom of the device so that the user can push the bell using the finger to touch the surface of the chest tightly. The diameter of the bell is chosen such that it is same as the standard size of an acoustic stethoscope 4.5 cm, to measure all ranges of physiological sounds. In addition, the user can check the heart rate and battery life of the device immediately on the organic light-emitting diode (OLED) display screen.

### 2.2. Embedded System Design of Proposed Smart Stethoscope

To achieve two core functions, namely heart sound and pulse measurement, two high-performance digital sensors, namely a micro-electromechanical systems (MEMS) digital microphone sensor and PPG sensor, were individually selected and designed in the proposed smart stethoscope. The selected microphone sensor (ICS-43434, TDK, Tokyo, Japan) consists of an MEMS sensor, signal conditioning, analog-to-digital (ADC) converter, decimation and antialiasing filters, and 24-bit I^2^S interface that allows the microphone to connect directly to the microprocessor without an audio codec to decrease data loss during extra processing. In addition, the microphone has a high signal-to-noise ratio (SNR) and wideband frequency response with sensitivity tolerance that is suitable for recording heart sounds because of its frequency range of 20–400 Hz [13]. A wearable health dedicated high-sensitivity pulse oximeter and heart-rate sensor (MAX30101, Maxim Integrated, San Jose, CA, USA) are used for pulse signal measurement because the sensor is integrated with pulse oximetry and a heart rate monitor module. They include internal LEDs, photodetectors, optical elements, and low-noise electronics with ambient light rejection. The sensor has a low-noise signal conditioning analog front-end with an 18-bit ADC, industry-lead ambient light cancellation circuit, and picket fence detection and replacement function. With a standard serial peripheral interface (SPI) compatible interface, the sensor can be directly connected to the microprocessor.

The proposed smart stethoscope is designed to measure heart sound and pulse simultaneously. In addition, the measured data must be transmitted to a smartphone in real time. Therefore, a multi-tasking support, high-performance, and low-power Bluetooth SoC ARM processor (EFR32BG, SiliconLabs, Austin, TX, USA) is designated as the main microprocessor. This chip includes an 80 MHz ARM Cortex-M33 core that provides a full digital signal processing instruction set and floating-point unit to rapidly process physiological signals. Its integrated security subsystem provides leading security features that significantly reduce the risk of IoT security breaches and compromised intellectual property to protect personal physiological data. Depending on the multi-tasking function, the heart sound and pulse signal processing are designed to be two parallel channels for collecting two signals in synchronization. In addition, a ping-pong buffer was designed for this system to process the heart sound continuously, from receiving sensor data to extracting the data features [11]. In addition, Bluetooth 5—LE long-range radio frequency (RF) link was designed in a microprocessor that provides reliable and compatible communications to connect the device and smartphone wirelessly for data transmission.

## 3. Material and Experiment

### 3.1. Circuit Design and PCB Manufacture

Figure 2 shows the proposed multimodal smart stethoscope PCB artwork design and manufactured PCB. The size of the PCB is designed to be 98 × 40 × 1.6 mm with a single-layer PCB structure. Except for the battery socket, the top surface consists of all the electrical elements of the sensor, power supply, and control communication parts, as shown in Figure 2a. Because the microphone sensor is a bottom port microphone, the microphone is placed on the upper of the PCB, and a small hole runs through the PCB that connects with the sensor directly to pass the sound vibration from the stethoscope bell. To improve the strength of wireless communication and decrease the dimensions of the PCB, an interdigital shape pattern antenna is designed for Bluetooth communication at 2.4 GHz frequency. A shading line was also intended to encase the RF elements that isolate the noise occurring from other electrical components to affect the wireless communication. At the bottom of the PCB, a 0.91-inch OLED dot-matrix graphic display screen consisting of 128 segments and 64 commons was selected and fixed on the PCB for pulse signals and heartbeats display. The display was driven by a single-chip CMOS OLED control driver and connected to a microprocessor unit via an inter-integrated circuit (I^2^C) interface. A 4.7 V 350 mAh Lithium recharge battery with a high-speed recharging circuit was designed for power supply. In addition, to increase the usage time, the operating voltage of the microprocessor and sensors was set to 1.8 V, which is the minimum value for most of the semiconductor elements. In addition, the USB recharging connector was upgraded to C type owing to its popularity and rich connection support for future improvement. The manufactured PCB is illustrated in Figure 2b.

### 3.2. Prototype Shape Manufacture and Assembly

The structure of the proposed multimodal smart stethoscope in a 3D design is shown in Figure 3. The assembly unit of the device was separated into four pieces: the bottom case, stethoscope bell diaphragm fastener, top case, and finger holder. All these units were manufactured using a high-quality 3D printer with acrylonitrile butadiene and styrene material and polished in post-processing for smooth texture of the inside and outside. The assembled smart stethoscope prototype is shown in Figure 4a, where the PPG sensor emits light on the finger and receives the reflected light. The housing of the PPG sensor is made of transparent glass with a thickness of 1 mm, and the gap between the housing and PPG sensor was only 1 mm, which is the optimal distance for signal measurement. Figure 4b shows that the bottom of the prototype consists of a stethoscope bell with a diaphragm and OLED display. A full-range stethoscope diaphragm (Spirit, Taipei, Taiwan) is used to cover the bell and is fixed on the device using a ring-shaped fastener with a rotary structure. Figure 4c,d show the front and lateral views, respectively, of a 160 cm tall female who grasps the device in the user mode. The figures indicate that the user can grab the device using a PC mouse, and the first segment of the middle finger can be attached to the PPG sensor housing in a position accurately guided by the finger holder.

### 3.3. Experiment to Test the Performance and Usability of the Proposed Smart Stethoscope

The assembled prototype of the proposed multimodal smart stethoscope was used to conduct an in vivo experiment that included two tasks: first, to verify the performance of the prototype in heart sound and pulse signal measurement in comparison with a professional physiological signals system, and second, to test the usability by evaluating the muscle fatigue level of the arm when using the prototype. Twenty healthy adult subjects (10 males and 10 females; age: 24–27 years old) were invited to participate in this study. The purpose and procedure of this study and experiment were informed to the subjects, and a consent was signed before the experiment. Before the experiment began, the subjects were required to sit comfortably on a chair and relax for approximately five minutes.

To verify the heart sound and pulse signals measured by the prototype and measure the electromyography (EMG) signal from the arm of the users for fatigue level evaluation, a multi-channel physiological data acquisition system MP160 (BIOPAC System, Inc., Goleta, CA, USA) with a contact acoustic transducer (SS17LA), PPG sensor (TSD200C), and EMG sensor were used for comparison with the prototype and to measure the EMG signals. As shown in Figure 5a, the acoustic transducer and PPG sensor are attached to the surface of the chest and middle finger of the left hand, respectively, to measure the heart sounds and pulse signals. The flexor digitorum profundus is a flexor of the wrist (midcarpal) and metacarpophalangeal and interphalangeal joints. It is used to aid the lumbrical muscles in their role as extensors of the interphalangeal joints, and power is transferred from the flexor digitorum profundus muscle to fully extend the fingers and flex the metacarpophalangeal joints. The electrodes of EMG sensor are attached to the inner part of the right arm to measure the EMG signal from flexor digitorum profundus. Meanwhile, the subject was asked to grasp the prototype with the right hand and attach it to the chest near the contact acoustic transducer during the experiment. Figure 5b shows the prototype being placed in different positions of the chest to observe the movement of the aortic valve (AV), tricuspid valve (TV), pulmonary valve (PV), and mitral valve (MV) and auscultate different heart diseases [14]. The experiment is conducted in four steps at four different positions on the chest in sequence at an interval of one minute.

Figure 5c,d show the second experiment conducted to evaluate the users’ muscle fatigue levels. The values obtained are compared for the proposed smart stethoscope and standard stethoscope. The subject was required to use the prototype and a standard stethoscope to measure the heart sound for 30 s each. Electrodes of the EMG sensor were attached to the outside of the right arm to measure superficial EMG signals. The data measured by the prototype and MP160 during these experiments were transmitted to a high-performance PC via Bluetooth and serial communication, respectively, for data storage and processing.

## 4. Results and Discussion

### 4.1. Heart Sound and Pulse Measurement Performance

Figure 6 shows an example of the experimental results of the performance test of the proposed smart stethoscope prototype for heart sound and pulse measurement. The heart sound and pulse signals were delineated as phonocardiogram (PCG) and PPG as shown in Figure 6a,b, respectively. The heart sound is measured from the aortic valve. The heart sounds S1 and S2 for the heart circle can be recognized easily in two PCG signals acquired by the prototype and MP160. The amplitude of the PCG signals measured by the prototype was lower than that measured by the acoustic transducer probably because the prototype measured heart sounds through clothing; however, the acoustic sensor measured the heart sounds on the skin of the chest directly. In addition, in the frequency domain, clothing blocks the high-frequency sounds that result in high-frequency components of heart sound measured by the prototype, which is less than that of the acoustic transducer. For pulse signal measurement, PPG signals measured by the two devices can be clearly identified with the peaks of the pulse. The signal measured by the prototype is more distinct and rounded than the signal measured by MP160, because the prototype includes baseline noise and residual noise reduction functions for preprocessing.

Table 1 shows the calculated cross-correlation value of the heart sound and PPG signals measured by the prototype and MP160 for 20 subjects. It can be observed that the mean values of heart sounds and PPG signals are approximately 1 with values of 0.958 ± 0.05 and 0.955 ± 0.04, respectively, indicating that the physiological signals measured by the prototype and MP160 are similar and positively correlated. The cross-correlation values of male subjects are slightly higher than those of female subjects. This is because the heart sound for females was measured on a cloth that was thicker than that of male subjects who were only wearing a shirt. The experimental results verified that the PCG and PPG signals delineated by the prototype were reliable.

### 4.2. Results of Usability Test

Figure 7 shows the examples of EMG signals measured for one subject during the two tasks of the experiment. Figure 7a shows the EMG signals measured from the flexor digitorum profundus while using the prototype to measure heart sounds in four positions: the AV, TV, PV, and MV in the first task. The EMG signal variation can be observed clearly when the subject is relaxed and moves the hand at four different positions on the chest during every heart sound measurement. The amplitude variation levels of the EMG signals at every heart sound measurement position were similar because the four positions on the surface of the chest were very close. Figure 7b shows the EMG signals measured at a superficial level, while the subject experimented with the second task for fatigue level evaluation using the prototype and standard stethoscope. The signal shows a clear difference in the time domain among large variations in the signal amplitude using the prototype and standard stethoscope. The experimental results show that the subject held the standard stethoscope harder than the prototype owing to the weight of the standard stethoscope (250 g), which was heavier than the prototype (85 g) by approximately three times.

In addition, the EMG signals measured from the two tasks were calculated with the root mean square (RMS) in time domain and average frequency (MEF) in frequency domain as the metrics to evaluate the potential of EMG in measuring subjects’ muscle fatigue [15]. The equations for calculating RMS and MEF values of the EMG signals are as follows:(1)$$\mathrm{RMS}=\sqrt{\frac{\sum_{i=1}^{N}{\mathrm{EMG}}^{2}}{N}}$$
(2)Power(EMG, f∈[0 Hz, 250 Hz])

Figure 8a,b show the RMS and MEF values for the flexor digitorum profundus muscle when the subjects used the prototype to measure heart sounds at four different positions in the first task. The results show that the values at all four positions are similar, proving that the subjects were using similar muscle fatigue. Figure 8c,d show the RMS and MEF value for the superficial muscle to compare the muscle fatigue when the subjects used the prototype and standard stethoscope. The results showed that the RMS value for the subject using a standard stethoscope was higher than that of the prototype, proving that the subject used more muscle fatigue while holding the prototype than when holding a standard stethoscope. In addition, there is a clear difference in the frequency domain that is used to calculate the MEF value between the two devices. The lower MEF values show greater muscle fatigue levels, which is in accordance with the results of the data calculated in the RMS value to verify that the subjects experience lower muscle fatigue while they held the prototype [16].

## 5. Conclusions

This study presented the design of a novel multimodal smart stethoscope for monitoring personal cardiovascular health status by simultaneously measuring the user’s heart sounds and pulse. Based on previous research, the shape of the stethoscope was designed to be similar to a compact PC mouse. The assembly was mounted on one PCB for easy grasp, and attachment to the chest surface using one hand helped to improve ruggedness and usability. A digital microphone and PPG sensor were equipped on the bottom and top surfaces of the device, respectively, to measure heart sounds and pulse variation from the user’s chest and finger simultaneously. In addition, to improve the signal processing capacity and convenience, a high-performance BLE Soc ARM microprocessor was implemented in the device for measured data pre-processing and communication with the smartphone. The prototype was assembled with a manufactured PCB and 3D-printed shell to conduct an in vivo experiment with 20 male and female subjects. The performance of heart sound and pulse signal measurement along with usability was measured by evaluating the users’ muscle fatigue levels. The experimental results showed that the prototype can measure heart sound and pulse signals simultaneously with high accuracy, which is similar to the professional physiological acquisition system. The experimental results also proved that the lightweight prototype decreases the fatigue level during heart sound measurement compared to standard stethoscopes in heart sound measurement. In future work, various cardiovascular disease patients will be asked to participate in the study to improve the performance of the device and to develop a cardiovascular disease self-diagnosis algorithm.

## Figures and Tables

**Figure 1 sensors-22-06465-f001:**
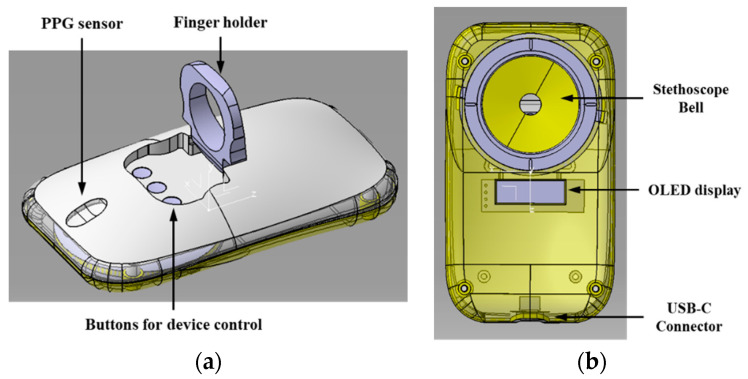
Design of the shape of proposed smart stethoscope: (**a**) top of the device and (**b**) bottom of the device.

**Figure 2 sensors-22-06465-f002:**
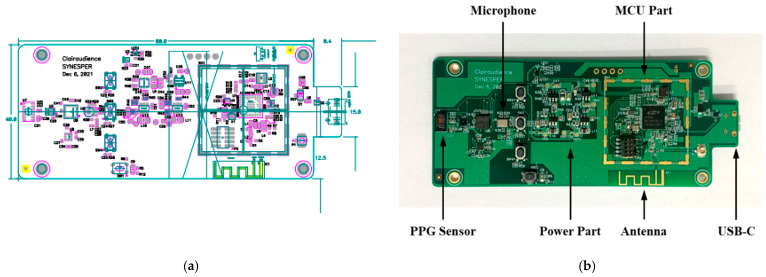
Proposed smart stethoscope PCB manufacture: (**a**) PCB artwork design; (**b**) Manufactured PCB.

**Figure 3 sensors-22-06465-f003:**
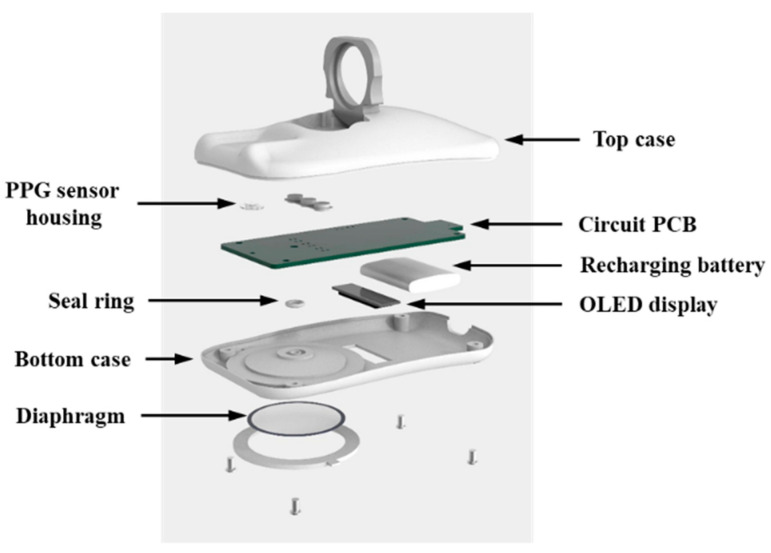
Structure of the proposed multimodal smart stethoscope in 3D.

**Figure 4 sensors-22-06465-f004:**
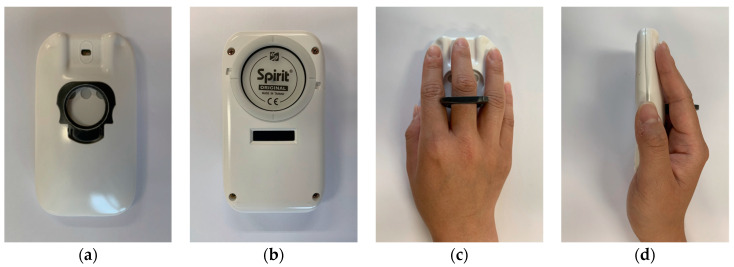
Pictures of assembled proposed multimodal smart stethoscope prototype: (**a**) top view of the prototype’s top; (**b**) top view of the prototype’s bottom; (**c**) top view of grasping the prototype; (**d**) side view of grasping the prototype.

**Figure 5 sensors-22-06465-f005:**
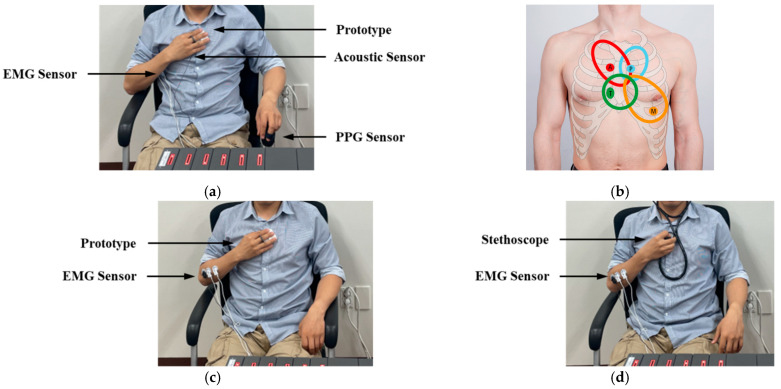
Pictures of the experimental setup for evaluating the performance and usability: (**a**) Photo of a subject in the performance test of physiological signal measurement; (**b**) auscultation positions; (**c**) EMG measurement with the prototype; (**d**) EMG measurement with a standard stethoscope.

**Figure 6 sensors-22-06465-f006:**
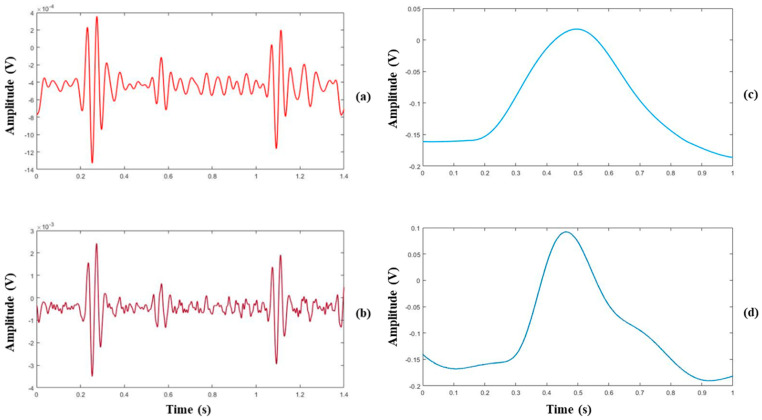
Example of heart sound (PCG) and pulse (PPG) signal measured by the prototype and MP160: (**a**) PCG signal delineated by the prototype device; (**b**) PCG signal delineated by MP160; (**c**) PPG signal delineated by the prototype; (**d**) PPG signal delineated by MP160.

**Figure 7 sensors-22-06465-f007:**
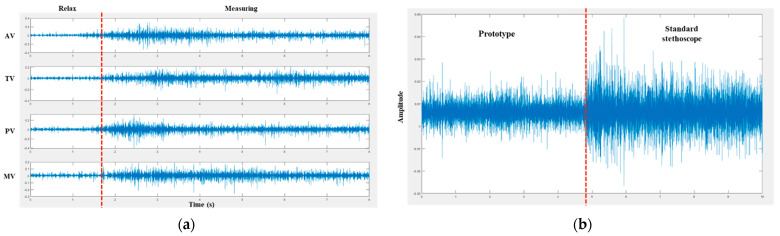
Example of EMG signals in performance and usability tests: (**a**) EMG signals for heart sound at four measurement positions; (**b**) EMG signals comparison between prototype and standard stethoscope.

**Figure 8 sensors-22-06465-f008:**
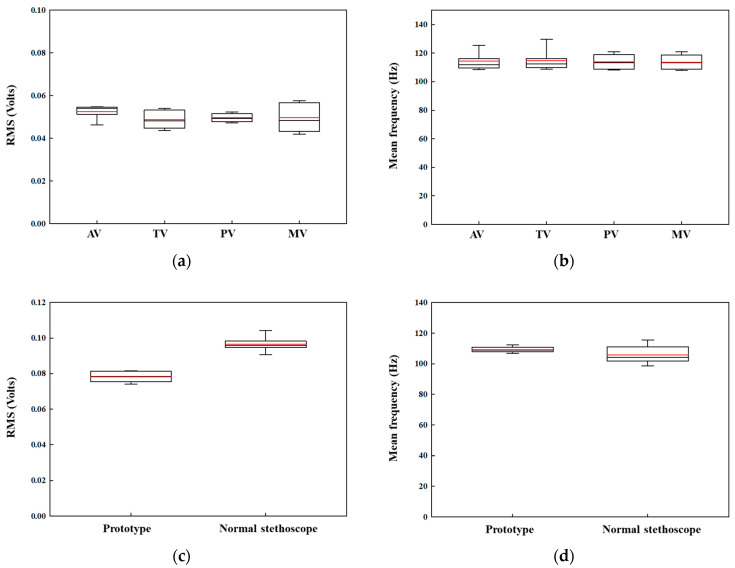
Values of RMS and MEF for EMG data were measured in two tasks: (**a**) RMS value for flexor digitorum profundus muscle in the first task; (**b**) MEF value for flexor digitorum profundus muscle in the first task; (**c**) RMS value for superficial muscle in the second task; (**d**) MEF value for superficial muscle in the second task.

**Table 1 sensors-22-06465-t001:** Cross-correlation value of PCG and PPG signal measured by prototype and MP160 for 20 subjects.

Subject Number	Gender	Heart Sounds		PPG Signals	
		Mean	Std.	Mean	Std.
Subject 1	M	0.97	0.04	0.95	0.05
Subject 2	M	0.98	0.03	0.96	0.03
Subject 3	M	0.96	0.04	0.96	0.07
Subject 4	M	0.97	0.04	0.95	0.03
Subject 5	M	0.97	0.06	0.97	0.03
Subject 6	M	0.98	0.02	0.95	0.05
Subject 7	M	0.95	0.07	0.94	0.07
Subject 8	M	0.98	0.04	0.97	0.04
Subject 9	M	0.97	0.03	0.96	0.07
Subject 10	M	0.96	0.05	0.95	0.02
Subject 11	F	0.96	0.09	0.95	0.05
Subject 12	F	0.94	0.08	0.98	0.01
Subject 13	F	0.93	0.07	0.96	0.04
Subject 14	F	0.95	0.07	0.95	0.03
Subject 15	F	0.94	0.09	0.94	0.06
Subject 16	F	0.97	0.06	0.95	0.05
Subject 17	F	0.93	0.07	0.94	0.04
Subject 18	F	0.96	0.02	0.95	0.09
Subject 19	F	0.93	0.06	0.96	0.03
Subject 20	F	0.96	0.06	0.97	0.03

## Data Availability

The dataset supporting the conclusions of this article is not available due to privacy and ethical reasons.

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
