# Peer review of "Novel Design of a Multimodal Technology-Based Smart Stethoscope for Personal Cardiovascular Health Monitoring"

_sensors, 2022, doi:10.3390/s22176465_

Round 1

Reviewer 1 Report

This paper proposed a novel smart stethoscope based on multimodal physiological signal measurement technology for personal cardiovascular health monitoring. The proposed device is designed in the shape of a compact personal computer mouse for easy grasping and attachment to the surface of the chest using only one hand, which is considered novel by reviewers. At the same time, the reviewer also pays attention to the following issues:

1. This paper developed a smart stethoscope that can be held by one hand to measure heart sounds and pulses simultaneously. Are pulse signals and heart sounds synchronized, or are they collected together but not completely synchronized? If it is synchronized, explain how it is done.

2. What is the sampling frequency of the Heart sound signals and pulse signals? what is the highest sampling frequency of the device?

3. For the signal quality of the prototype designed in this paper, only the average amplitude is compared. The envelope shape of the pulse signal in Figure 6(c) is roughly the same as that in Figure 6(d), but the details of the descending part do not seem to be exactly the same, but that part contains some important information. Please explain it.

Reviewer 2 Report

The authors presented a novel smart stethoscope based on multimodal physiological signal measurement technology for personal cardiovascular health monitoring. The work is interesting and meaningful.
However, there are two major problems in this work that need further consideration:
1. Why only do the in-vivo experiment with 20 young adults? What is the audience for which this device was invented?
2. Why is the test instrument invented in this paper not compared with the results of other existing similar instruments?

Round 2

Reviewer 2 Report

I have no more questions.